# Temperature dependence of adsorption hysteresis in flexible metal organic frameworks

Shamsur Rahman[1], Arash Arami-Niya[1,2], Xiaoxian Yang[1], Gongkui Xiao [1], Gang (Kevin) Li[1,3] & Eric F. May [1✉]

"Breathing" and "gating" are striking phenomena exhibited by flexible metal-organic frameworks (MOFs) in which their pore structures transform upon external stimuli. These effects are often associated with eminent steps and hysteresis in sorption isotherms. Despite significant mechanistic studies, the accurate description of stepped isotherms and hysteresis remains a barrier to the promised applications of flexible MOFs in molecular sieving, storage and sensing. Here, we investigate the temperature dependence of structural transformations in three flexible MOFs and present a new isotherm model to consistently analyse the transition pressures and step widths. The transition pressure reduces exponentially with decreasing temperature as does the degree of hysteresis (c.f. capillary condensation). The MOF structural transition enthalpies range from $+6$ to $+31\,\mathrm{kJ \cdot mol^{-1}}$ revealing that the adsorption-triggered transition is entropically driven. Pressure swing adsorption process simulations based on flexible MOFs that utilise the model reveal how isotherm hysteresis can affect separation performance.

[1] Fluid Science & Resources Division, School of Engineering, University of Western Australia, Crawley, WA 6009, Australia. [2] Discipline of Chemical Engineering, Western Australian School of Mines: Minerals, Energy and Chemical Engineering, Curtin University, GPO Box U1987Perth, WA 6845, Australia. [3] Department of Chemical Engineering, The University of Melbourne, Parkville, VIC 3010, Australia. ✉email: eric.may@uwa.edu.au

The development and study of flexible porous materials, and metal–organic frameworks (MOFs) in particular, has been a major focus of the adsorption community over recent years. Interest in these materials can be attributed to their potential use in a wide range of applications for gas separation, gas storage, and catalysis[1–8]. The host–guest interaction in flexible MOFs affects the host's pore characteristics and the associated accessibility by the guest molecule. The interaction may be classified as either: (1) a crystallographic phase transition causing a unit cell volume change (breathing effect), or (2) a linker rotation and sub-net sliding with no change in the unit cell volume (gate opening). In both cases, the ability of these "soft materials" to change structure means their observed adsorption characteristics are fundamentally different to that of conventional adsorbents[9–12]. The most obvious difference is the drastic increase in uptake that occurs when the number of accessible adsorption sites in the framework suddenly changes in response to external stimuli, such as the adsorption or desorption of guest molecules[13–15]. This feature of flexible MOFs allows the adsorption of guest molecules seemingly larger than the nominal crystallographic pore diameter to suddenly increase. Such materials offer, in principle, a very high selectivity and could be very useful in the development of gas separation technologies[16]. However, studies of MOF-based applications are limited by the lack of isotherm models able to accurately describe their abrupt changes in sorption capacities.

Structural transitions triggered by gas-phase pressure are guest-dependent phenomena that cause the MOF to change from a nonporous (np) phase (narrow pores in breathing MOFs or closed pores in gating MOFs) to a porous (lp) phase (expanded, large pores in breathing MOFs or open pores in gating MOFs)[13,14,17–19]. This structural change occurs when a critical transition pressure, $p_{tr}$, is reached, causing a step change in the isotherm characterized by a finite transition width, $\sigma$ (in Pa). Moreover, the phase change is hysteretic[20–22] with the transition pressure measured along the adsorption branch of the isotherm (increasing pressure), $p_{tr}^{(ads)}$, being greater than that measured along the desorption branch, $p_{tr}^{(des)}$, with the difference $\delta p_{tr} \equiv (p_{tr}^{(ads)} - p_{tr}^{(des)})$ quantifying the degree of hysteresis typically being significantly larger than $\sigma$. In this work, we investigated the dependence of $\delta p_{tr}$ on temperature for three different MOFs and explored their asymptotic behavior at low temperatures.

Fundamental simulations capable of modeling responsive adsorption processes in flexible porous materials have been developed[23]. These simulations combine DFT calculations of adsorption in slit pores with Helmholtz energy descriptions of the crystal as function of pore width, and cannot be readily fit to experimentally measured isotherms for real MOF samples, or used in simulations of pressure swing adsorption processes based on such materials. Here, we utilize a new empirical model for flexible adsorbents that can efficiently extract salient features of measured hysteretic MOF sorption isotherms.

The new empirical isotherm model is used to quantify $p_{tr}$, $\delta p_{tr}$, and $\sigma$ and their temperature dependences for three MOFs by regression to uptake data both newly measured here and taken from the literature. Sorption measurements of $CO_2$ on ZIF-7 down to 233 K reveal that, for flexible MOFs, the isotherm hysteresis reaches an asymptotic limit with decreasing temperature. We also show the new model more accurately represents the separation performance achievable in pressure swing adsorption processes based on flexible MOFs.

## Results

The wide-ranging data of Mason et al.[2] for methane sorption on iron(II)-1,4-benzenedipyrazolate (Fe(bdp)) and Cobalt-1,4-benzenedipyrazolate (Co(bdp)) provide excellent examples of MOF hysteretic stepped sorption isotherms. Figure 1 shows the $CH_4$ sorption capacity data along the two extreme isotherms[2] reported for each MOF. The number of data acquired along each branch of each hysteretic sorption isotherm is sufficient to clearly resolve the characteristic parameters $p_{tr}$, $\delta p_{tr}$, and $\sigma$, as well as their respective temperature dependences.

To extract these parameters from stepped sorption isotherms, we developed an empirical model (Equation 1(a)) that can be regressed to capacity data, $q$, measured along each branch, the basis and derivation of which is detailed in Note 1 of the Supplementary Information (SI).

$$q = q_{LJMY} + q_{Langmuir}, \quad (1a)$$

$$q_{LJMY} = \frac{Q_{step}}{2}\left[1 + erf\left(\frac{p - p_{tr}}{\sqrt{2}\sigma}\right)\right], \quad (1b)$$

$$q_{Langmuir} = Q_m \frac{Kp}{1 + Kp}. \quad (1c)$$

Here, $q_{Langmuir}$ is a Langmuir isotherm function (Equation 1(c)) with two adjustable parameters, $K$ and $Q_m$, characterizing the initial slope and limiting sorption amount, respectively, at pressures below the np $\leftrightarrow$ lp phase transition. The function $q_{LJMY}$ (Equation 1(b)) describes the stepped part of the sorption isotherm using three additional adjustable parameters, $p_{tr}$, $\sigma$, and $Q_{step}$, where the latter represents the increase in capacity after the phase transition (above that of $q_{Langmuir}$). As detailed in Note 1 of the SI, the LJMY model can be derived by assuming the sorption site energies made available by the MOF's structural phase transition have a Gaussian distribution. It is analogous to the LJM model developed by Li et al.[24], which describes the temperature-regulated admission and release of gases in microporous trapdoor materials. To describe a branch of a hysteretic stepped sorption

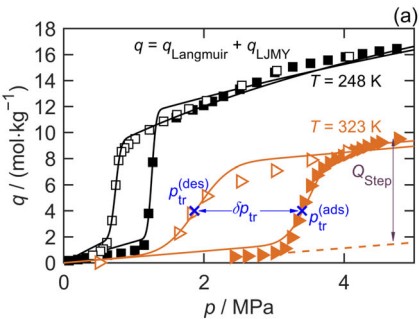
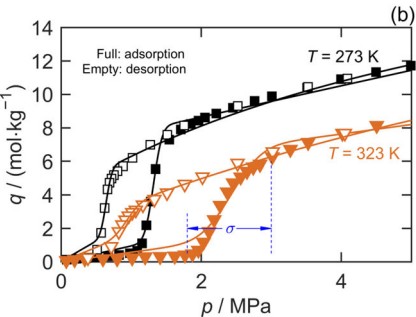

**Fig. 1 Exemplar hysteretic sorption isotherms for CH₄ on the MOFs measured by Mason et al.[2], illustrating the key parameters in Equation 1. a** Fe(bdp) and **b** Co(bdp). Full symbols denote the adsorption branch and hollow symbols denote the desorption branch. Squares denote the lower temperature isotherm and triangles the higher temperature.

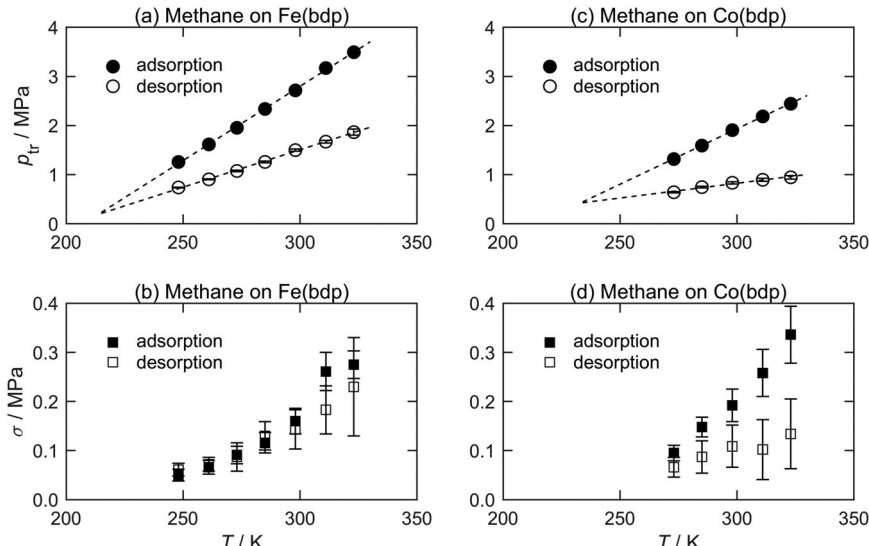

**Fig. 2 Transition pressure and transition width vs temperature for the sorption of $CH_4$ on MOFs measured by Mason et al.[2].** Circles denote transition pressures, $p_{tr}$, for **a** Fe(bdp) and **c** Co(bdp), while squares denote transition widths, $\sigma$, for **b** Fe(bdp) and **d** Co(bdp). Full symbols denote parameters from the adsorption branch and hollow symbols denote parameters from the desorption branch. Statistical uncertainties are shown as error bars in all panels, but are smaller than the symbol for $p_{tr}$ in **a** and **c**.

isotherm for a given MOF, the LJMY model may be combined with any classical sorption isotherm model (e.g., Sips and Toth), as required to best describe the data.

Equation 1 was regressed to each branch of the sorption isotherms for Fe(bdp) and Co(bdp) measured by Mason et al[2]. to determine the five parameters, $p_{tr}$, $\sigma$, $Q_{step}$, $K$, and $Q_m$. The best-fit LJMY parameter values, their statistical uncertainties and the fit's standard error are listed for $CH_4$ sorption on Fe(bdp) and Co(bdp) in Supplementary Tables 1 and 2, respectively (Note 2 of the SI). Figure 2 shows the temperature dependence and hysteresis of the best-fit parameters $p_{tr}$ and $\sigma$ for these MOFs. Figure 2 (a) and (c) shows that the transition pressures measured on the adsorption and desorption branches have statistically significant differences in their temperature dependence, with $p_{tr}^{(ads)}$ varying more rapidly in both cases than $p_{tr}^{(des)}$. In contrast, the degree of hysteresis in the transition width is much smaller, with no statistically significant difference between $\sigma^{(ads)}$ and $\sigma^{(des)}$ observed for Fe(bdp) at any temperature. For Co(bdp), a small difference in the temperature dependence of $\sigma^{(ads)}$ and $\sigma^{(des)}$ is apparent, although at 323 K the (largest) difference is only about twice the combined statistical uncertainties of the fit parameters.

Figure 2 shows that the width of the MOF's structural hysteresis, $\delta p_{tr}$, decreases with reducing temperature. This is opposite to the behavior observed for hysteresis associated with capillary condensation in classical adsorbents with a rigid structure, which becomes larger at lower temperatures[25,26].

Simple extrapolations of the apparently linear trends in $p_{tr}^{(ads)}$ and $p_{tr}^{(des)}$ with temperature exhibited in Figure 2 suggest that the hysteresis in the sorption isotherms might disappear (e.g., $\delta p_{tr} \rightarrow 0$) around 210 and 230 K for Fe(bdp) and Co(bdp), respectively. Measuring sorption capacities accurately at such temperatures is very challenging and so an alternative flexible adsorbent was sought to investigate the asymptotic behavior of structural hysteresis with temperature. Analysis of the data published by Arami-Niya et al.[27] for $CO_2$ sorption on a zeolitic imidazolate framework (ZIF-7) above 273 K suggested that $\delta p_{tr} \rightarrow 0$ as $T \rightarrow$ 265 K, a significantly more accessible temperature.

Accordingly, ZIF-7 was synthesized following Arami-Niya et al.[27]. The adsorption and desorption of pure $CO_2$ on the ZIF-7 was measured at eight temperatures between 233 and 293 K at

pressures up to 0.1 MPa. Figure 3 shows the measured data together with the fits of the LJMY-Langmuir model for each isotherm; these are also listed in Supplementary Table 3 (fit parameters) and Supplementary Table 5 (data). At the lowest temperatures measured, ZIF-7 exists in the lp phase even when the $CO_2$ pressure is about 2 kPa. Moreover the hysteresis for $CO_2$ on ZIF-7 becomes progressively smaller as the temperature is reduced, consistent with the observations of $\delta p_{tr}$ for Fe(bdp) and Co(bdp). However, $\delta p_{tr}$ does not become zero at any of the temperatures measured for ZIF-7.

At the four lowest temperatures shown in Figure 3 for $CO_2$ on ZIF-7, $\delta p_{tr}$ and $\sigma$ are both <3 kPa. This makes high resolution measurements of the step in the sorption isotherm difficult and determination of $p_{tr}$, $\sigma$, $Q_{step}$, $K$, and $Q_m$ by regressing each branch separately can be challenging. In such cases, simultaneously fitting both branches of the hysteretic sorption isotherm as described in the Note 1 of the SI, and shown in Supplementary Figures 1–3 can be helpful.

Figure 4 shows that over a wide temperature range the transition pressures do not vary linearly, but are instead described by the Clausius–Clapeyron equation.

$$\ln\left(\frac{p_{tr}}{p_{tr_0}}\right) = \frac{\Delta H_{tr}}{R}\left[\frac{1}{T_0} - \frac{1}{T}\right]. \qquad (2)$$

Here, $p_{tr_0}$ is the transition pressure (for either adsorption or desorption), at a reference temperature, $T_0 = 273.15$ K; $R$ is the universal gas constant; and $\Delta H_{tr}$ is the enthalpy associated with the material's structural transition (np $\leftrightarrow$ lp). Regression of Equation 2 to the measured $p_{tr}^{(ads)}$ for $CO_2$ on ZIF-7 gives $\Delta H_{tr}^{(ads)} = (30.1 \pm 1.1)$ kJ mol$^{-1}$, while fitting to $p_{tr}^{(des)}$ gives $\Delta H_{tr}^{(des)} = (31.4 \pm 1.6)$ kJ mol$^{-1}$. Thus, while there is an offset between the transition pressures measured in adsorption and desorption, (reflected by different values of $p_{tr_0}$ for the two branches) their dependence on temperature is the same. However, over the temperature ranges measured by Mason et al.[2], $\Delta H_{tr}^{(ads)} = (9.1 \pm 0.2)$ kJ mol$^{-1}$ and $\Delta H_{tr}^{(des)} = (8.3 \pm 0.1)$ kJ mol$^{-1}$ for $CH_4$ on Fe(bdp), while $\Delta H_{tr}^{(ads)} = (9.1 \pm 0.3)$ kJ mol$^{-1}$ and $\Delta H_{tr}^{(des)} = (5.6 \pm 0.5)$ kJ mol$^{-1}$ for $CH_4$ on Co(bdp). Supplementary Table 4 (Note 2 of the SI) presents the results of fits to isotherms measured for $CO_2$ on amino-MIL-53 (Al)[28], from

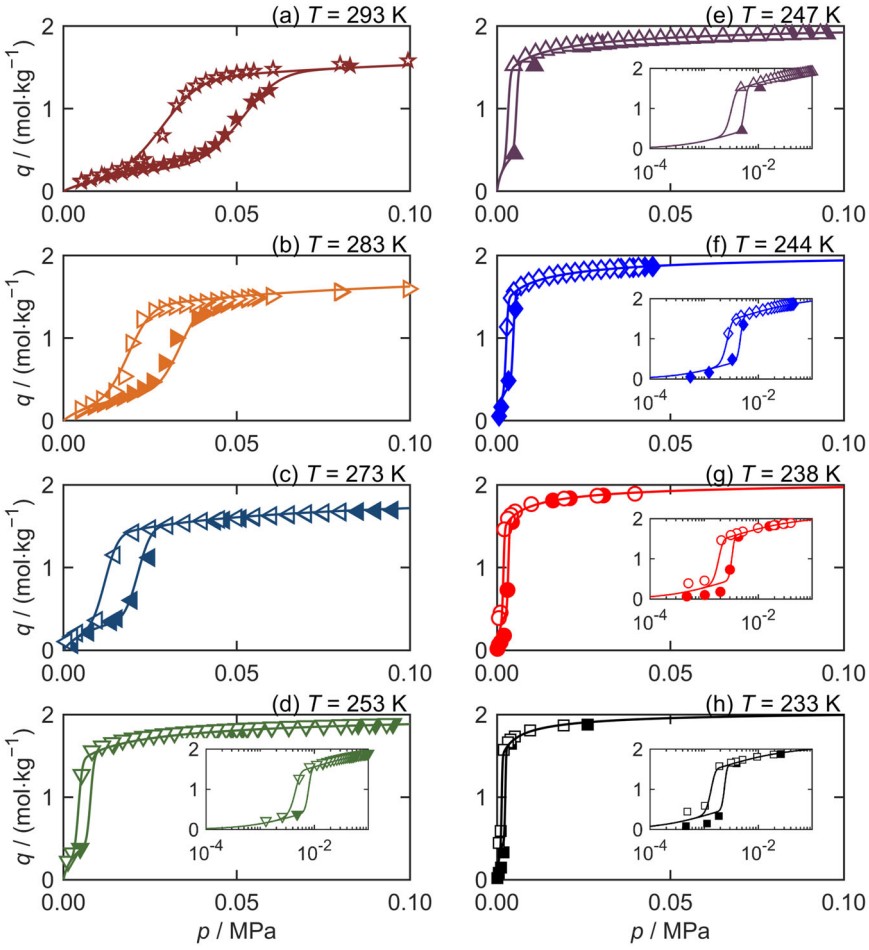

**Fig. 3 Asymptotic isotherm hysteresis with decreasing temperature for CO₂ on ZIF-7.** Equilibrium $CO_2$ capacities, $q$, measured in adsorption (filled symbols) and desorption (empty symbols) for ZIF-7 at pressures, $p$, to 0.1 MPa, for eight temperatures $T$ from **a** 293 K to **h** 233 K. Curves represent fits of the LJMY-Langmuir model (Equation 1 and described in Note 1 of the SI). Insets show the hysteresis of low temperature isotherms on a logarithmic pressure scale.

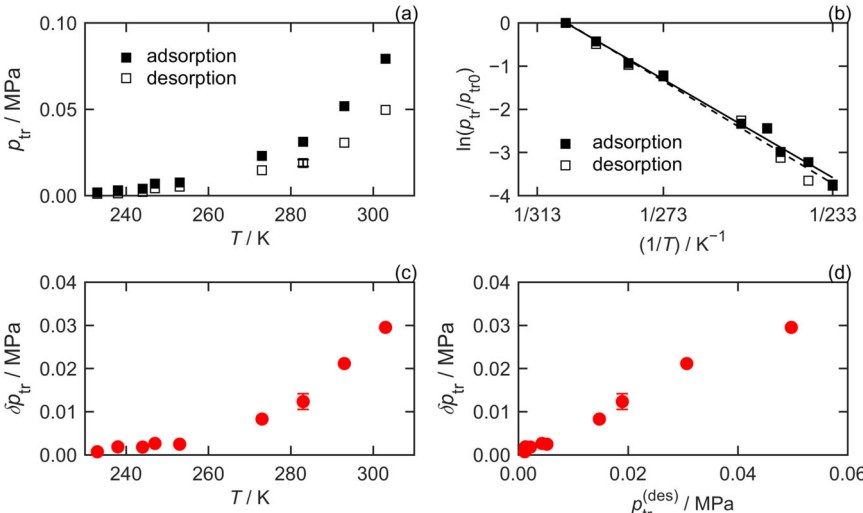

**Fig. 4 Limiting behavior of transition pressures and hysteresis parameters.** Squares denote transition pressures, $p_{tr}$, measured in adsorption (solid symbols) and desorption (hollow symbols), as a function of **a** temperature, $T$ or **b** relative to the transition pressure at 303 K, $p_{tr0}$, for CO₂ on ZIF-7. Circles denote the hysteresis of the isotherm's transition pressure, $\delta p_{tr}$, with **c** temperature and **d** with the transition pressure measured in desorption, $p_{tr}^{(des)}$.

which values of $\Delta H_{tr}^{(ads)} = 15\,kJ\,mol^{-1}$ and $\Delta H_{tr}^{(des)} = 19\,kJ\,mol^{-1}$ can be estimated. In general, the temperature dependence of $p_{tr}^{(ads)}$ and $p_{tr}^{(des)}$ could depend on the MOF, adsorbate, and/or the temperature range considered.

The asymptotic temperature dependence of $\delta p_{tr}$ follows from Equation 2. Wherever $\delta H_{tr} \equiv \Delta H_{tr}^{(ads)} - \Delta H_{tr}^{(des)}$ is finite, the transition pressure hysteresis will vary according to

$$\delta p_{tr} = \left( \frac{p_{tr_0}^{(ads)}}{p_{tr_0}^{(des)}} \exp\left( \frac{\delta H_{tr}}{R}\left[ \frac{1}{T_0} - \frac{1}{T} \right] \right) - 1 \right) p_{tr}^{(des)}, \qquad (3)$$

where the temperature dependence of $p_{tr}^{(des)}$ is given by Equation 2. When $\delta H_{tr} = 0$ (e.g., $CO_2$ on ZIF-7)

$$\delta p_{tr} = \delta p_{tr_0} \exp\left( \frac{\Delta H_{tr}}{R}\left[ \frac{1}{T_0} - \frac{1}{T} \right] \right), \qquad (4a)$$

$$\delta p_{tr} = \left( \frac{p_{tr_0}^{(ads)}}{p_{tr_0}^{(des)}} - 1 \right) p_{tr}^{(des)}. \qquad (4b)$$

Here, $\delta p_{tr_0}$ is the difference in the transition pressures on the two branches at the reference temperature. Equation 4(a) explains the exponential trend shown in Figure 4(c) and Equation 4(b) explains the linear correlation in Figure 4(d). Equations 3 and 4 also enable the asymptotic behavior of the transition pressure hysteresis observed for Fe(bdp) and Co(bdp) to be predicted reliably, in contrast to the simplistic linear extrapolations with temperature shown in Figure 2.

## Discussion

As discussed by Schneeman et al.[14], the hysteresis observed in a MOF's stepped isotherm is due to the energy penalty associated with the increasing interfacial area as np → lp. No such energy barrier needs to be overcome for the reverse lp → np structural change: hence the isotherm's desorption branch corresponds to thermodynamic equilibrium, while the adsorption branch reflects the system accessing metastable states. The positive $\Delta H_{tr}$ values indicate that the MOF's structural transition is entropically driven with the lp state being less-ordered than the np state. This is consistent with previous but less precise estimates by microcalorimetry: for $CO_2$-driven structural transitions, Llewellyn et al.[29] estimated $+20\,kJ\,mol^{-1}$ for MIL-53 (Cr) and Du et al.[30] estimated $+7\,kJ\,mol^{-1}$ for ZIF-7. The latter is significantly smaller than the value obtained here likely because of the difficulty in separating the exothermic adsorption process from the endothermic structural transition. Interpreting the nonzero value of $\delta H_{tr} \equiv \Delta H_{tr}^{(ads)} - \Delta H_{tr}^{(des)}$ observed for $CH_4$ on Co(bdp) is more difficult. Potentially, nonzero $\delta H_{tr}$ values might reflect a combination of effects (e.g., structural transition plus pore filling/emptying, hysteresis in crystal lattice strain) occurring simultaneously and/or asymmetrically along the isotherm's two branches. Further discussion on the temperature dependence and hysteresis of the structural transition parameters can be found in Note 4 of the SI. Supplementary Figure 4 shows the various extents of hysteresis observed in the transition width parameter as a function of temperature for $CH_4$ on Fe(bdp), $CH_4$ on Co(bdp), and $CO_2$ on ZIF-7.

Note 5 of the SI shows how the LJMY-Langmuir model can be used to more reliably simulate PVSA separation processes utilizing flexible MOFs. To achieve optimal separation performance, the transition pressures and widths on both branches of the isotherm must be considered appropriately, when specifying the PVSA cycle's high and low pressures. In addition, use of only a single branch (desorption or adsorption) in the simulation will produce an overly optimistic prediction of the separation performance, relative to that calculated with the fully hysteretic isotherm (see Supplementary Figures 5–7 and Supplementary Tables 6–8).

## Conclusions

The temperature dependence and hysteresis of stepped sorption isotherms associated with structural transitions was studied for three flexible MOFs: Fe(bdp) with $CH_4$, Co(bdp) with $CH_4$, and ZIF-7 with $CO_2$. A five-parameter model developed to describe stepped sorption isotherms enabled the transition pressures, $p_{tr}$, widths, $\sigma$, and hysteresis $\delta p_{tr} \equiv \left( p_{tr}^{(ads)} - p_{tr}^{(des)} \right)$, to be consistently and robustly analyzed. In contrast to capillary condensation, $\delta p_{tr}$ increases with temperature with a Clausius–Clapeyron dependence. At low temperatures, $\delta p_{tr}$ approaches an asymptotic limit proportional to $p_{tr}^{(des)}$. The MOFs' structural phase transitions are entropically driven with enthalpy changes ranging from 5 to 3 kJ $mol^{-1}$. Curiously, the structural transition enthalpy for $CH_4$ on Co(bdp) is 60% larger in adsorption than desorption, possibly due to multiple effects occurring simultaneously or asymmetrically along the two isotherm branches.

## Methods

**Gas adsorption measurements**. A volumetric measurement system (model ASAP2020 by Micromeritics) was used to measure ZIF-7 sorption capacity for $CO_2$ along multiple isotherms[31]. Before the isotherm measurements, the ZIF-7 sample[27] was thoroughly dehydrated and degassed by heating stepwise to 473 K under high vacuum overnight. The degassed sample was cooled to room temperature and backfilled with helium. Adsorption isotherms for $CO_2$ on ZIF-7 were measured in the temperature range of 233–293 K at pressures up to 120 kPa. The adsorption temperatures were controlled by a homemade heating jacket connected to a thermostatic liquid bath (Ultra-Low Refrigerated Circulators JULABO FP88) filled with either silicone oil (for $T > 283$ K) or ethanol ($T \leq 283$ K). The temperature of the bath fluid was measured by a 100 Ω platinum resistance thermometer.

The equilibration required at each pressure was determined automatically by the ASAP2020 system with the criterion being a relative pressure change of <0.01% over an interval of 15 s. For the adsorption measurements on ZIF-7, the equilibration time varied from 5 min to 15 h across all temperatures, with an average value of 92 min. For the desorption measurements, the equilibration time ranged from 5 min to ~4 days, with an average value of 2 h. The longest equilibration times occurred at low temperatures (233 and 238 K) and pressures less than ~1 kPa. The relative combined standard uncertainty of sorption capacity, $u_c(q_i)/q_i$, measured with this ASAP2020 was estimated previously[31,32] to be 1.4%.

## Data availability

The data that support the findings of this study are available in the Supplementary Information or from the corresponding author on reasonable request.

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

## Acknowledgements
This research was funded by the Australian Research Council through IC150100019 and DP190100983. S.R. was supported by an Australian Government Research Training Program Scholarship.

## Author contributions
S.R., A.A.-N., G.L., and E.F.M. conceived and designed the experiments. S.R. and A.A.-N. performed the synthesis and measured the adsorption isotherms. S.R., A.A.-N., and E.F.M. interpreted the adsorption data and discussed the findings in this paper. X.Y., G.L., and E.F.M. derived the stepped adsorption isotherms model. G.X. and S.R. performed the Aspen Adsorption simulations. S.R., A.A.-N., G.L., and E.F.M. wrote the paper. All authors discussed the results and commented on the manuscript.

## Competing interests
The authors declare no competing interests.
