## [Peer Review File · Communications Chemistry]

Reviewers' comments:

Reviewer #1 (Remarks to the Author):

In this contribution, the authors hypothesize a new isotherm model that accounts for the hysteresis observed in MOFs by considering a distribution of local adsorption sites and associated transition pressures. This isotherm model describes the change in capacity upon the structural transition via three parameters, and should be combined with a traditional isotherm model such as the Langmuir isotherm to obtain a complete isotherm model. First, the authors show the validity of the model by regressing to published data for Fe(bdp) and Co(bdp), after which the authors perform new carbon dioxide sorption measurements on ZIF-7. The fits to these new data are then used to discuss the temperature dependence of the transition pressure and hysteresis.

Overall, this manuscript provides an interesting point of view regarding sorption in MOFs and therefore deserves publication. However, at this point, there are several open questions regarding the validity and physical interpretation of the model, as noted below. I would encourage the authors to incorporate these suggestions to further improve their manuscript before publication.

Major remarks

1. The proposed isotherm builds on the sensible hypothesis that variations in the local pore environment and adsorption sites of the MOF result in a distribution of transition pressures rather than a single well-defined transition pressure, therefore allowing certain regions of the material to undergo a transition before other regions. This, however, will also result in a strained framework, as the np and lp phases often do not exhibit the same unit cell parameters. Is this strain incorporated in the current model and, if not, to what extent do the authors think this would affect their model?
2. As the authors note, the model isotherm can be used to regress both the adsorption and desorption branches simultaneously, ensuring that four out of five parameters for both branches are the same. This is also what one would expect physically, as for instance the total adsorption capacity is the same in both branches. However, from Tables S1 and S2, it is clear that the σ and, more importantly, the Q_{step} parameter both depend on whether the adsorption or desorption branch are considered. For Fe(bdp) at 323 K, for instance, Q_{step} varies between 3.9 mol/kg for desorption and 8.01 mol/kg for adsorption, more than three times the standard error on either of the values and therefore significant. Do the authors have a physical explanation for this? Does it make sense to fit the adsorption and desorption branches separately? What does the model predict if both branches are fitted simultaneously? This sensitivity seems important to investigate before drawing conclusions from the model.
3. In continuation of the point above, Tables S1 and following should also include the fitted values of K and Q_m of the Langmuir model, as also here one would expect that these values are the same in both the adsorption and desorption branches.
4. For methane adsorption in Co(bdp), the authors observe that the σ parameter depends on whether one considers the adsorption or the desorption branch. Following the physical origin of the isotherm, this would mean that the variety of pore environments is different during adsorption and desorption. Can the authors explain this difference?
5. In Table S3, it is unclear which data are measured in this work and which are reproduced from earlier work in Ref. 4.
6. What is the uncertainty on the enthalpy associated with the structural transitions in MIL-53(Al) observed by Coucke et al.? In the text, only the fitted values are given.
7. Why is the nonzero value of ΔH_{tr} observed for methane adsorption on Co(bdp) and Fe(bdp)

more difficult to explain than the nonzero value for MIL-53(Cr)?

8. In the introduction, the authors classify the host-guest interactions in flexible MOFs as interactions that either cause a crystallographic phase transition and unit cell volume change (breathing or swelling) or result in linker rotations and sub-net sliding. However, in the case of swelling, such as in MIL-88, the phase of the material does not change, contrary to this statement, and there are no new interaction sites that are made available upon guest adsorption and no appreciable hysteresis. Given the scope of this article, it is therefore advisable to treat this swelling effect separately from the breathing effect (or even omit it completely). Would it make sense to use the here derived model for swelling MOFs?

9. What was the typical equilibration time to obtain the equilibrium capacities of Table S5? Do the authors have an idea about the uncertainty on these values?

Minor remarks

10. On a very minor note: the authors discuss in their article structural transition induced by gas adsorption or desorption, as opposed to transitions that take place under the stimulus of temperature, pressure, or other effects. The first sentence of the second paragraph on page 2 ("Pressure-dependent structural transitions...") may in this sense be misinterpreted as discussing structural transitions that are induced by a mechanical pressure rather than gas pressure; it might be better to slightly rephrase this.

11. It might be advisable to add to the caption of Figure 2 that error bars are present in all panels, as the error bars in panels (a) and (c) are very small.

12. On the last paragraph of page 5, the data of the experiments with ZIF-7 are collected in Table S5, not Table S4 as stated in the text.

13. The authors mention in the introduction that the MOF structural transition enthalpies range from 6 to 30 kJ/mol; however, in the conclusions, a range of 5 to 31 kJ/mol is mentioned.

14. On line 56 of the SI, the reference to eq. (S1a) should probably be replaced by a reference to eq. (S3).

Reviewer #2 (Remarks to the Author):

The manuscript from Rahman et al. describes their investigation of adsorption hysteresis in flexible metal-organic framework materials. The hysteresis and temperature dependence of three, previously reported, stepped sorption isotherms were studied: CH₄ on Fe(bdp), CH₄ on Co(bdp) and CO₂ on ZIF-7 with CO₂. The authors fit an empirical model to the isotherm by employing a five-parameter model, which combined a Langmuir model and a function to describe the step. As noted by the authors, this step function is analogous to a previously reported model developed to describe temperature-regulated gas adsorption. This empirical model was fit to the adsorption and desorption branch of isotherms to give robust parameters of transition pressures and hysteresis. It is observed that the transition pressure hysteresis increases with temperature following a Clausius-Clapeyron dependence. Subsequently, the derived empirical models can be used to reliably simulate pressure vacuum swing adsorption (PVSA) separation processes

[11]

The manuscript demonstrates an interesting and well-described model to analyse stepped isotherms and appears to extract meaningful information for the industrial application of these processes. I believe the quality of this work is commensurate with the quality expected by readers of Communications Chemistry but I have apprehension whether the work is of general interest to the chemistry community. It is without a doubt of interest to the adsorption community. This report is publishable subject to revision, following the comments raised below:

1. The LJM-Langmuir model is the key result from this report and the regression for treating the Fe(bdp) and Co(bdp) isotherms should also be graphically reported (as they have for ZIF-7, Figure 3). As this allows the reader to quickly observe how well the raw data is reproduced by this model.
2. This model relies upon Langmuir adsorption processes for the two adsorption states, can this model be extended to treat flexible mesoporous materials? What are some of the limitations of this model?
2. As you have demonstrated the adsorption enthalpy for Co(bdp) is very different in the adsorption and desorption branch. There are several reports of the pressure and temperature dependence of these values, especially for flexible mesoporous materials. How would this affect your models for transition pressure hysteresis?
3. The temperature dependence of transition pressures for Fe(bdp) and Co(bdp) does not show the asymptotic behaviour as it does in ZIF-7. Could this be related to the very different flexible behaviour responsible for the stepped isotherms? Do you believe the LJM-Langmuir is robust to treat the variety of different stepped isotherms reported in the literature (swelling, negative gas adsorption etc.)?
4. The temperature dependence of another anomalous adsorption process was recently discussed by Krause et al. [10.1039/D0FD00013B] They showed non-monotonic behaviour was observed upon temperature decrease. This may suggest that prediction using this empirical model could be limited to specific combinations of temperature, guest and material?

Response to Reviews of COMMSCHEM-20-0259-T

Title: Temperature Dependence of Adsorption Hysteresis in Flexible Metal Organic Frameworks

We thank the reviewers for their commentary on our submission, and for their suggestions on possible improvements. Below, the original comments of the reviewers appear in black Calibri font with our replies shown in blue Calibri font. Manuscript text is shown in black Times New Roman font, with changes to the manuscript shown in red Times New Roman font. All page numbers refer to the revised manuscript. In the following, we give a point-by-point reply to the reviewers' comments:

Reviewer 1

In this contribution, the authors hypothesize a new isotherm model that accounts for the hysteresis observed in MOFs by considering a distribution of local adsorption sites and associated transition pressures. This isotherm model describes the change in capacity upon the structural transition via three parameters, and should be combined with a traditional isotherm model such as the Langmuir isotherm to obtain a complete isotherm model. First, the authors show the validity of the model by regressing to published data for Fe(bdp) and Co(bdp), after which the authors perform new carbon dioxide sorption measurements on ZIF-7. The fits to these new data are then used to discuss the temperature dependence of the transition pressure and hysteresis.

Overall, this manuscript provides an interesting point of view regarding sorption in MOFs and therefore deserves publication. However, at this point, there are several open questions regarding the validity and physical interpretation of the model, as noted below. I would encourage the authors to incorporate these suggestions to further improve their manuscript before publication.

Major remarks

1. The proposed isotherm builds on the sensible hypothesis that variations in the local pore environment and adsorption sites of the MOF result in a distribution of transition pressures rather than a single well-defined transition pressure, therefore allowing certain regions of the material to undergo a transition before other regions. This, however, will also result in a strained framework, as the np and lp phases often do not exhibit the same unit cell parameters. Is this strain incorporated in the current model and, if not, to what extent do the authors think this would affect their model?

A strained framework would likely exist during the transition but this is not incorporated within the empirical model, which simply aims to correlate the observed equilibrium sorption capacities as a function of pressure and temperature. The model does not make reference to unit cell parameters for the np and lp phases, and as such it can not integrate the effect of strain on either the location of the structural transition or the amount adsorbed. The empirical model simply parameterizes the consequent features of the sorption isotherm allowing them, and their dependence on temperature, to be quantified.

Fundamental models capable of modelling responsive adsorption processes in flexible porous materials have been developed, which relate the change in pore width to the sorption capacity;

however these models use DFT to represent adsorption in slit pores and can not be readily fit to experimentally measured isotherms for real samples. The parameters extracted by fitting the LJMY isotherm to experimental data could in principle be related to the predictions of fundamental models that couple pore width with DFT calculations and can account for the effect of strain in the crystal. This might provide a pathway to efficiently quantifying the effect of strain on observed isotherms.

Of the parameters in the LJMY model, the transition width, σ , is the most likely to be influenced by the effects of strain, particularly in those cases where there is hysteresis in its value between the adsorption and desorption branch. For example differences in strain caused by the np to lp transition from those caused by the lp to np transition might manifest themselves in a non-zero $\delta\sigma = \sigma^{(ads)} - \sigma^{(des)}$ as observed for Co(bdp).

To cover these points, we added the following text to the Introduction of the manuscript:

Fundamental simulations capable of modelling responsive adsorption processes in flexible porous materials have been developed.¹ These simulations combine DFT calculations of adsorption in slit pores with Helmholtz energy descriptions of the crystal as function of pore width, and cannot be readily fit to experimentally-measured isotherms for real MOF samples, or used in simulations of pressure swing adsorption processes based on such materials. Here we utilise a new empirical model for flexible adsorbents that can efficiently extract salient features of measured hysteretic MOF sorption isotherms and evaluate their dependence on temperature.

and the following text to the first section of the SI:

Fundamental simulations capable of modelling responsive adsorption processes in flexible porous materials have been developed¹, which allow for a strained framework to exist during the transition between the np and lp phases as the unit cell parameters change.^{2,3} The parameters extracted by fitting the LJMY isotherm to experimental data could in principle be related to the predictions of fundamental models that couple pore width with DFT calculations and can account for the effect of strain in the crystal. This might provide a pathway to efficiently quantifying the effect of strain on observed isotherms.

Of the parameters in the LJMY model, the transition width, σ , is the most likely to be influenced by the effects of strain, particularly in those cases where there is hysteresis in its value between the adsorption and desorption branch. For example differences in strain caused by the np to lp transition from those caused by the lp to np transition might manifest themselves in a non-zero $\delta\sigma = \sigma^{(ads)} - \sigma^{(des)}$ as observed for Co(bdp).

2. As the authors note, the model isotherm can be used to regress both the adsorption and desorption branches simultaneously, ensuring that four out of five parameters for both branches are the same. This is also what one would expect physically, as for instance the total adsorption capacity is the same in both branches. However, from Tables S1 and S2, it is clear that the σ and, more importantly, the Q_{step} parameter both depend on whether the adsorption or desorption

branch are considered. For Fe(bdp) at 323 K, for instance, Q_{step} varies between 3.9 mol/kg for desorption and 8.01 mol/kg for adsorption, more than three times the standard error on either of the values and therefore significant. Do the authors have a physical explanation for this? Does it make sense to fit the adsorption and desorption branches separately? What does the model predict if both branches are fitted simultaneously? This sensitivity seems important to investigate before drawing conclusions from the model.

We have included in Section 2 of the SI the results of fitting the isotherm branches simultaneously. The best fit parameters and statistical uncertainties are included in Tables S1 to S3, and the results of the simultaneous fits to the data are shown in the new Figures S1 to S3. The following text was added to Section 2 of the SI.

Tables S1 to S3 show the best-fit parameters obtained from regressing the LJMY-Langmuir model to the MOF isotherm data. For each isotherm, three sets of parameters are listed: a fit to each branch separately and the results of fitting both branches simultaneously with eq (S3).

Fitting both branches simultaneously with eq (S3) constrains σ , Q_{step} , Q_m and K to have the same value in adsorption and desorption, preventing any investigation of hysteresis in these quantities. Particularly for the Fe(bdp) and Co(bdp) data, different values of Q_{step} on the two isotherm branches are needed to accurately represent the measured data across all conditions, especially when the hysteresis in transition pressure becomes at higher temperatures. Appreciably worse fits are achieved in these cases if the branches are fit simultaneously.

A possible physical explanation for the hysteresis in Q_{step} suggested by fitting the branches separately is that at higher temperatures the properties of the adsorbed phase do not lower the MOF's interfacial free energy as effectively and hence significantly more adsorption is necessary to induce the np to lp structural transition.

However, care should be taken not to over-interpret the best-fit value of Q_{step} obtained from a fit to a single isotherm branch, because it is correlated with Q_m and K_p . In contrast, $p^{(\text{tr})}$ and σ , which are of primary interest in this work, are robustly orthogonal to the other parameters in the model; their values and uncertainties do not change significantly if the branches are fit simultaneously, although their statistical uncertainties inherently increase if the fit quality deteriorates. Accordingly to more accurately investigate the dependence of $p^{(\text{tr})}$ and σ on temperature and isotherm branch, the values obtained by fitting the independent branches were used in the main text.

3. In continuation of the point above, Tables S1 and following should also include the fitted values of K and Q_m of the Langmuir model, as also here one would expect that these values are the same in both the adsorption and desorption branches.

The values of K and Q_m from the Langmuir component of the model were added to Tables S1 to S4, as requested by the reviewer.

4. For methane adsorption in Co(bdp), the authors observe that the σ parameter depends on whether one considers the adsorption or the desorption branch. Following the physical origin of the isotherm, this would mean that the variety of pore environments is different during adsorption and desorption. Can the authors explain this difference?

The hysteresis of the σ parameter with isotherm branch was discussed in Section 4 of the SI. We can not provide a definitive explanation for the hysteresis observed for Co(bdp) but can speculate as to the cause, particularly in light of the response to the Reviewer's first point. We have added the following text to Section 4 of the SI.

Collectively, the results from these three MOFs suggest there might be a temperature below which σ is the same for both branches of the isotherm, while at higher temperatures a hysteresis in this parameter becomes increasingly manifest. As suggested in Section 1, differences in the value of σ obtained for each branch might reflect differences in the crystal strain caused by the direction of the np-lp transition. Alternatively (or in addition) the magnitude and temperature dependence of σ might contain information about the sample's heterogeneity. Clearly, however, such speculation needs to be investigated further by measuring hysteretic sorption isotherms over a wider range of T , with additional MOF-adsorbate combinations, and through fundamental simulations with methods such as those of Evans et al.¹

5. In Table S3, it is unclear which data are measured in this work and which are reproduced from earlier work in Ref. 4.

As per the reviewer's suggestion, the source of the data is now specified in the caption of Table S3:

Table S3. Values and statistical standard uncertainties, u , of best fit parameters p_{tr} , σ , & Q_{step} determined by regression of the LJMY-Langmuir model (eq (1) or eq (S3)) to the data measured in this work (233 to 293 K) and by Arami-Niya et al.⁴ (303 K) for the sorption of CO₂ on ZIF-7, together with the standard error of fit.

6. What is the uncertainty on the enthalpy associated with the structural transitions in MIL-53(Al) observed by Coucke et al.? In the text, only the fitted values are given.

Couck et al. only reported sorption isotherms measured two temperatures. Information from two temperatures is sufficient to estimate the enthalpy associated with the structural transitions in MIL-53(Al) but it is not possible to determine the statistical uncertainty of that estimate from only two points.

7. Why is the nonzero value of ΔH_{tr} observed for methane adsorption on Co(bdp) and Fe(bdp) more difficult to explain than the nonzero value for MIL-53(Cr)?

We believe the notation used in the text was somewhat confusing and have modified it to improve clarity. Non-zero values of ΔH_{tr} are not difficult to explain for any MOF; this is the enthalpy

associated with the **np-lp** transition. What is difficult to explain solely from the experimental data analysed in this work is a non-zero value of $\delta H_{tr} \equiv \Delta H_{tr}^{(ads)} - \Delta H_{tr}^{(des)}$. We have clarified the text as follows:

Interpreting the non-zero value of $\delta H_{tr} \equiv \Delta H_{tr}^{(ads)} - \Delta H_{tr}^{(des)}$ observed for CH₄ on Co(bdp) is more difficult. Potentially, non-zero δH_{tr} values might reflect a combination of effects (e.g. structural transition plus pore filling/emptying, **hysteresis in crystal lattice strain**) occurring simultaneously and/or asymmetrically along the isotherm's two branches.

We now refer only to CH₄ on Co(bdp) because this is the only data set considered for which a finite δH_{tr} is statistically significant.

8. In the introduction, the authors classify the host-guest interactions in flexible MOFs as interactions that either cause a crystallographic phase transition and unit cell volume change (breathing or swelling) or result in linker rotations and sub-net sliding. However, in the case of swelling, such as in MIL-88, the phase of the material does not change, contrary to this statement, and there are no new interaction sites that are made available upon guest adsorption and no appreciable hysteresis. Given the scope of this article, it is therefore advisable to treat this swelling effect separately from the breathing effect (or even omit it completely). Would it make sense to use the here derived model for swelling MOFs?

We agree with the reviewer and have removed the reference to swelling from the Introduction. We now refer only to breathing and gate-opening.

9. What was the typical equilibration time to obtain the equilibrium capacities of Table S5? Do the authors have an idea about the uncertainty on these values?

We have added the following information about the equilibration time and sorption capacity uncertainties to the Methods section of the main text.

The equilibration required at each pressure was determined automatically by the ASAP2020 system with the criterion being a relative pressure change of less than 0.01% over an interval of 15 seconds. For the adsorption measurements on ZIF-7, the equilibration time varied from 5 minutes to 15 hours across all temperatures, with an average value of 92 minutes. For the desorption measurements, the equilibration time ranged from 5 minutes to about 4 days, with an average value of 2 hours. The longest equilibration times occurred at low temperatures (233 K and 238 K) and pressures less than about 1 kPa. The relative combined standard uncertainty of sorption capacity, $u_c(q_i)/q_i$, measured with this ASAP2020 was estimated previously^{4,5} to be 1.4 %.

We also now indicate the relative uncertainty of the ZIF-7 sorption capacity measurements in the

caption of Table S5 in the SI.

Minor remarks

10. On a very minor note: the authors discuss in their article structural transition induced by gas adsorption or desorption, as opposed to transitions that take place under the stimulus of temperature, pressure, or other effects. The first sentence of the second paragraph on page 2 (“Pressure-dependent structural transitions...”) may in this sense be misinterpreted as discussing structural transitions that are induced by a mechanical pressure rather than gas pressure; it might be better to slightly rephrase this.

The text was revised as per the reviewer’s suggestion:

“Structural transitions triggered by gas phase pressure are guest-dependent phenomena that cause the MOF to change from a non-porous (np) phase (narrow-pores in breathing MOFs or closed-pores in gating MOFs) to a porous (lp) phase (expanded, large pores in breathing MOFs or open-pores in gating MOFs).”

11. It might be advisable to add to the caption of Figure 2 that error bars are present in all panels, as the error bars in panels (a) and (c) are very small.

As suggested, “Statistical uncertainties are shown as error bars in all panels but are smaller than the symbol for p_{tr} in (a) and (c).” was added at the end of the caption of Figure 2.

12. On the last paragraph of page 5, the data of the experiments with ZIF-7 are collected in Table S5, not Table S4 as stated in the text.

The text was revised as directed.

13. The authors mention in the introduction that the MOF structural transition enthalpies range from 6 to 30 kJ/mol; however, in the conclusions, a range of 5 to 31 kJ/mol is mentioned.

The text was revised in the abstract:

“The MOF structural transition enthalpies range from +6 to +31 kJ·mol⁻¹ revealing that ...”

14. On line 56 of the SI, the reference to eq. (S1a) should probably be replaced by a reference to eq. (S3).

The reference was intended to be to eq (1a) of the main text, where the Langmuir and LJM models are combined additively. This has been corrected.

Reviewer #2:

The manuscript from Rahman et al. describes their investigation of adsorption hysteresis in flexible metal-organic framework materials. The hysteresis and temperature dependence of three, previously reported, stepped sorption isotherms were studied: CH₄ on Fe(bdp), CH₄ on Co(bdp) and CO₂ on ZIF-7 with CO₂. The authors fit an empirical model to the isotherm by employing a five-parameter model, which combined a Langmuir model and a function to describe the step. As noted by the authors, this step function is analogous to a previously reported model developed to describe temperature-regulated gas adsorption. This empirical model was fit to the adsorption and desorption branch of isotherms to give robust parameters of transition pressures and hysteresis. It is observed that the transition pressure hysteresis increases with temperature following a Clausius-Clapeyron dependence. Subsequently, the derived empirical models can be used to reliably simulate pressure vacuum swing adsorption (PVSA) separation processes

The manuscript demonstrates an interesting and well-described model to analyse stepped isotherms and appears to extract meaningful information for the industrial application of these processes. I believe the quality of this work is commensurate with the quality expected by readers of Communications Chemistry but I have apprehension whether the work is of general interest to the chemistry community. It is without a doubt of interest to the adsorption community. This report is publishable subject to revision, following the comments raised below:

1. The LJM-Langmuir model is the key result from this report and the regression for treating the Fe(bdp) and Co(bdp) isotherms should also be graphically reported (as they have for ZIF-7, Figure 3). As this allows the reader to quickly observe how well the raw data is reproduced by this model.

As suggested, we added Figures S1 and S2 to the SI, which show the model fits to the CH₄ adsorption data on Fe(bdp) and Co(bdp), respectively.

2. This model relies upon Langmuir adsorption processes for the two adsorption states, can this model be extended to treat flexible mesoporous materials? What are some of the limitations of this model?

The proposed model is empirical and provides a useful framework for quantitatively characterizing sorption phenomena observed in flexible materials. However, it does not necessarily provide an explanation for those phenomena. Modifications to the manuscript and the Supplementary Information made in response to Reviewer 1's similar comments (see above) now make this point about the model's limitations more clearly.

3. As you have demonstrated the adsorption enthalpy for Co(bdp) is very different in the adsorption and desorption branch. There are several reports of the pressure and temperature dependence of these values, especially for flexible mesoporous materials. How would this affect your models for transition pressure hysteresis?

We demonstrate that the enthalpy of the structural transition for Co(bdp) is different on the isotherm branches; this is a different enthalpy to that associated with the heat of adsorption on the two branches. While the effect of temperature on the adsorption enthalpy has been reported for several flexible materials, here we present the first (to our knowledge) analysis of structural transition enthalpies in such materials. We note in the first paragraph of the Discussion section (page 9) that direct measurements of structural transition enthalpies by calorimetry are difficult because they can not be separated from the heat of adsorption signal.

Transition pressure hysteresis (i.e. different values of p_{tr} on the two branches) reflects the energy penalty associated with the MOFs' structural transition that must be paid in adsorption (the metastable state) vs the equilibrium state observed in desorption. Further explanation is given in the first paragraph of the Discussion (page 9). Different structural transition enthalpies on the two branches, which are estimated from differences in the temperature dependences of $p_{tr}^{(ads)}$ and $p_{tr}^{(des)}$, might reflect differences in crystallographic strain induced by the transformation, with more work needing to be done in one direction than the other. The manuscript now mentions this point in the Discussion section.

Interpreting the non-zero value of $\delta H_{tr} \equiv \Delta H_{tr}^{(ads)} - \Delta H_{tr}^{(des)}$ observed for CH₄ on Co(bdp) is more difficult. Potentially, non-zero δH_{tr} values might reflect a combination of effects (e.g. structural transition plus pore filling/emptying, **hysteresis in crystal lattice strain**) occurring simultaneously and/or asymmetrically along the isotherm's two branches.

4. The temperature dependence of transition pressures for Fe(bdp) and Co(bdp) does not show the asymptotic behaviour as it does in ZIF-7. Could this be related to the very different flexible behaviour responsible for the stepped isotherms?

This is a key point of the manuscript: equations (3) and (4) allow the asymptotic behavior of the transition pressures with temperature to be evaluated for all three MOFs. ZIF-7 and Fe(bdp) are described by eq (4) because these materials have $\delta H_{tr} \equiv \Delta H_{tr}^{(ads)} - \Delta H_{tr}^{(des)} \approx 0$ for methane, while the asymptotic behavior of Co(bdp) is described by eq (3) because $\delta H_{tr} \approx 4.5 \text{ kJ.mol}^{-1}$. This is in contrast to the inferences drawn from simple extrapolations of the apparently linear trends in $p_{tr}^{(ads)}$ and $p_{tr}^{(des)}$ with temperature exhibited in Figure 2. To clarify this point, the following text has been added to the end of the Results section following eqs (3) and (4).

Equation (**Error! Reference source not found.**) explains the exponential trend shown in Figure 4c

and the linear correlation in Figure 4d. Equations (3) and (4) also enable the asymptotic behavior of the transition pressure hysteresis observed for Fe(bdp) and Co(bdp) to be predicted reliably, in contrast to the simplistic linear extrapolations with temperature shown in Figure 2.

Do you believe the LJM-Y-Langmuir is robust to treat the variety of different stepped isotherms reported in the literature (swelling, negative gas adsorption etc.)?

The LJM-Y-Langmuir model can not treat all stepped isotherms reported in the literature, such as negative gas adsorption where the observed sorption capacity does not increase monotonically with pressure. This is discussed further in response to Reviewer 2's final comment below. We have removed any reference in the manuscript to swelling transitions as also suggested by Reviewer 1.

4. The temperature dependence of another anomalous adsorption process was recently discussed by Krause et al. [10.1039/D0FD00013B] They showed non-monotonic behaviour was observed upon temperature decrease. This may suggest that prediction using this empirical model could be limited to specific combinations of temperature, guest and material?

We thank the reviewer for raising this point and now cite the important work of Krause et al. The empirical model is a useful tool for analyzing some classes of stepped isotherms exhibited by specific combinations of temperature, guest and material and representing them in PSA process simulations based on such combinations. This is now made clear in the final paragraphs of Section 1 of the SI.

However, the LJM-Y model does not make any direct reference to crystal strain. Any connection between the LJM-Y parameters extracted by fitting to sorption isotherm data and more fundamental simulations that do include crystal strain will need to be inferred by analyzing the predictions of those simulations. Additionally, the LJM-Y model can not treat stepped isotherms where the observed sorption capacity does not increase monotonically with pressure, for example as observed for materials that exhibit negative gas adsorption². The primary purpose of the LJM-Y model is to serve as a useful tool for analyzing experimental sorption isotherms measured for particular combinations of guests and flexible adsorbents, and representing those isotherms in simulations of pressure swing adsorption processes based on those materials.

References

- 1 Evans, J. D., Krause, S., Kaskel, S., Sweatman, M. B. & Sarkisov, L. Exploring the thermodynamic criteria for responsive adsorption processes. *Chemical Science* **10**, 5011-5017, doi:10.1039/C9SC01299K (2019).
- 2 Krause, S. et al. The Role of Temperature and Adsorbate on Negative Gas Adsorption in the Mesoporous Metal-Organic Framework DUT-49. *Faraday Discussions*, doi:10.1039/D0FD00013B (2020).
- 3 Neimark, A. V., Coudert, F.-X., Boutin, A. & Fuchs, A. H. Stress-Based Model for the Breathing of Metal-Organic Frameworks. *The Journal of Physical Chemistry Letters* **1**, 445-449, doi:10.1021/jz9003087 (2010).
- 4 Wedler, C. et al. Gas Diffusion and Sorption in Carbon Conversion. *Energy Procedia* **158**, 1792-1797, doi:<https://doi.org/10.1016/j.egypro.2019.01.422> (2019).

- 5 Xiao, G., Li, Z., Saleman, T. L. & May, E. F. Adsorption equilibria and kinetics of CH₄ and N₂ on commercial zeolites and carbons. *Adsorption* **23**, 131-147, doi:10.1007/s10450-016-9840-7 (2017).

REVIEWERS' COMMENTS:

Reviewer #1 (Remarks to the Author):

I thank the authors for taking into account the suggestions raised during the previous reviewing round, thereby substantially improving and clarifying their manuscript. I'm therefore happy to recommend the current version of the manuscript for publication, although I have one remaining suggestion the authors may want to take into account.

To my understanding, the parameters K and Q_m of the Langmuir model lose their usual interpretation, as discussed on page 4 of the manuscript, when combined with the LJM isotherm model, given the correlation between these parameters and Q_{step} . This for instance explains the different K parameters in adsorption and desorption, although they would have the same physical interpretation in the Langmuir model. Is this correct? In that case, while the isotherm model suggested here is still very valuable to compare different isotherms, I would deem it necessary to explicitly mention this, as it would be incorrect to simply extract the K and Q_m parameters from this model and simply report these as sometimes done with the standard Langmuir model.

Reviewer #2 (Remarks to the Author):

Since my previous review, it appears the authors have made considerable amendments to the manuscript and addressed the critiques raised by the reviewers. The response to the reviewers' comments is reflected in the manuscript. I believe this work is suitable for publication in Communications Chemistry.

Response to Reviews of COMMSCHEM-20-0259A

Title: Temperature Dependence of Adsorption Hysteresis in Flexible Metal Organic Frameworks

We thank the reviewers for their commentary on our submission, and for their suggestions on possible improvements. Below, the original comments of the reviewers appear in black Calibri font with our replies shown in blue Calibri font. Manuscript text is shown in black Times New Roman font, with changes to the manuscript shown in red Times New Roman font. All page numbers refer to the revised manuscript. In the following, we give a point-by-point reply to the reviewers' comments:

Reviewer 1

I thank the authors for taking into account the suggestions raised during the previous reviewing round, thereby substantially improving and clarifying their manuscript. I'm therefore happy to recommend the current version of the manuscript for publication, although I have one remaining suggestion the authors may want to take into account.

To my understanding, the parameters K and Q_m of the Langmuir model lose their usual interpretation, as discussed on page 4 of the manuscript, when combined with the LJM isotherm model, given the correlation between these parameters and Q_{step} . This for instance explains the different K parameters in adsorption and desorption, although they would have the same physical interpretation in the Langmuir model. Is this correct? In that case, while the isotherm model suggested here is still very valuable to compare different isotherms, I would deem it necessary to explicitly mention this, as it would be incorrect to simply extract the K and Q_m parameters from this model and simply report these as sometimes done with the standard Langmuir model.

We agree with the Reviewer regarding this point and have added an additional paragraph to page 4 of the SI:

Additionally, the potential for correlation between Q_{step} , Q_m and K means that values of the latter two Langmuir parameters should be interpreted with caution. This is particularly important for fits where there are a relatively small number of data points measured at pressures below the structural transition, and/or where the equilibrium capacity of the adsorbent at $p < p_{tr}$. Under these conditions, the statistical uncertainties associated with Q_m and K can be large even if the fit quality for the entire isotherm is good. For example, different values of Q_m and K obtained by fitting separately to the adsorption and desorption branches of an isotherm might simply reflect the larger number of data points measurable at $p < p_{tr}^{(ads)}$ during adsorption than at $p < p_{tr}^{(des)}$ during desorption.